# Genome-Wide Identification, Evolution, and Expression Analysis of *TPS* and *TPP* Gene Families in *Brachypodium distachyon*

**DOI:** 10.3390/plants8100362

**Published:** 2019-09-23

**Authors:** Song Wang, Kai Ouyang, Kai Wang

**Affiliations:** 1Key Laboratory of Genetics, Breeding and Multiple Utilization of Crops, Ministry of Education, Fujian Provincial Key Laboratory of Haixia Applied Plant Systems Biology, Fujian Agriculture and Forestry University, Fuzhou 350002, Fujian, China; 2National Engineering Research Center of Sugarcane, Fujian Agriculture and Forestry University, Fuzhou 350002, Fujian, China; 3Key Laboratory of Sugarcane Biology and Genetic Breeding, Ministry of Agriculture and Rural Affairs, Fujian Agriculture and Forestry University, Fuzhou 350002, Fujian, China

**Keywords:** TPS, TPP, *Brachypodium distachyon*, abiotic stress, expression analysis

## Abstract

Trehalose biosynthesis enzyme homologues in plants contain two families, trehalose-6-phosphate synthases (TPSs) and trehalose-6-phosphate phosphatases (TPPs). Both families participate in trehalose synthesis and a variety of stress-resistance processes. Here, nine *BdTPS* and ten *BdTPP* genes were identified based on the *Brachypodium distachyon* genome, and all genes were classified into three classes. The Class I and Class II members differed substantially in gene structures, conserved motifs, and protein sequence identities, implying varied gene functions. Gene duplication analysis showed that one *BdTPS* gene pair and four *BdTPP* gene pairs are formed by duplication events. The value of *Ka/Ks* (non-synonymous/synonymous) was less than 1, suggesting purifying selection in these gene families. The *cis*-elements and gene interaction network prediction showed that many family members may be involved in stress responses. The quantitative real-time reverse transcription (qRT-PCR) results further supported that most *BdTPSs* responded to at least one stress or abscisic acid (ABA) treatment, whereas over half of *BdTPPs* were downregulated after stress treatment, implying that *BdTPSs* play a more important role in stress responses than *BdTPPs*. This work provides a foundation for the genome-wide identification of the *B. distachyon TPS–TPP* gene families and a frame for further studies of these gene families in abiotic stress responses.

## 1. Introduction 

Abiotic stress, i.e., drought, chilling, salinity, and heat stress, are limiting factors that seriously affect crop quality and yield; thus, improving the abiotic stress tolerance of plants is highly important for the agricultural industry. Plants have gradually evolved complex molecular mechanisms that respond to environmental stress, resulting in stress tolerance. Complex regulatory networks can cause certain molecular, biochemical, and physiological changes, including the accumulation of osmotic substances, stomatal closure, and decreased photosynthesis [1]. Among these changes, the accumulation of trehalose is also a significant pathway for plants under stress conditions to avoid damage.

Trehalose is a non-reducing disaccharide of glucose that increases under abiotic stress in organisms [2]. Previous studies have found trehalose, for instance, to participate in the responses of microorganism, especially bacteria and yeast to different abiotic stresses [3,4,5,6]. In some microorganisms and invertebrate animals, trehalose acts as a carbon source and an osmoprotectant [7,8], particularly under heat and drought stress [9]. Indeed, some studies have shown that trehalose can protect plant cytomembranes against both drought injuries and freezing injuries [10] and maintain the stability of enzymes [11]. In higher plants, trehalose was determined to provide protection under different abiotic stresses, i.e., drought, heat, chilling, salinity, etc. [12,13,14,15]. 

It has previously been reported that trehalose can be biosynthesized via more than five routes in prokaryotes, but only one of these pathways exists in plants, namely, the ‘TPS–TPP pathway’ [2]. This route includes two enzymatic steps: The first process is catalyzed by trehalose-6-phosphate synthase (TPS) and synthesizes an intermediate from UDP-glucose (UDPG) and glucose 6-phosphate (G6P); next, trehalose-6-phosphate phosphatase (TPP) dephosphorylates the intermediate to trehalose [2,16]. The TPS protein has been previously identified in many organisms [17,18,19,20], such as *Saccharomyces cerevisiae* and *Escherichia coli.* In recent years, *TPS* and *TPP* genes have been studied in many higher plants, suggesting that trehalose is likely ubiquitous in flowering plants [2,8,21,22,23,24]. Unlike single *TPS* and *TPP* genes in the majority of microorganisms, the genomes of higher plants encode a large family of trehalose biosynthesis enzyme homologues. These members are commonly classified into three classes based on the TPS/TPP protein domain [25,26]. In the *Arabidopsis* genome, four Class I (*AtTPS1–4*), seven Class II (*AtTPS5–AtTPS11*), and ten Class III (*AtTPPA–AtTPPJ*) genes have been identified [24,27].

In higher plants, trehalose-6-phosphate (T6P) is an indispensable signal intermediate that influences plant growth and development by linking pivotal signaling routes of carbon metabolism [16,24,28,29]. First, T6P plays a vital role in starch synthesis via the post-translational redox activation of ADP-glucose pyrophosphorylase [30,31]. Second, T6P can inhibit the activity of Sn1-related protein kinase (SnRK1) [32], which has a vital role in the transcription networks of plant stress and energy metabolism [33]. From a review of the literature, the genetic manipulation of *TPS/TPP* genes could improve stress resistance in different species [15,34,35,36,37,38,39]. Moreover, some *TPS/TPP* genes are directly involved in stress tolerance by improving the trehalose contents in several plants [14,40,41]. However, many of the functions of the *TPS/TPP* genes are largely unknown, especially for those genes involved in signaling pathways in plant development and stress resistance. Therefore, the identification of *TPS* and *TPP* genes is crucial for investigating the molecular mechanisms that combat various environmental stresses.

*Brachypodium distachyon L*. (2n = 10), as a model system for grasses, has a small genome (~270 Mbp), low plant height, and a short growth cycle [42,43], which enables the evaluation of the molecular mechanisms of growth, development, and abiotic stress. Although *TPS* genes in other plants were identified [27,44,45,46,47], *TPS* genes in *B. distachyon* have not been well studied. Moreover, the *TPP* family has not yet been characterized in plants other than *Arabidopsis* [24]. In this study, the characteristics of the *BdTPS* and *BdTPP* genes were identified by analyzing the evolutionary tree, intron/exon structure, motifs, and selective forces. We also studied the expression patterns in different tissues and under various stresses. Taken together, our study enhances the understanding of the function and evolution of the *TPS* and *TPP* genes.

## 2. Results

### 2.1. Identification of TPS and TPP Family Members in Brachypodium distachyon

To identify all homologs of TPS and TPP in *B. distachyon*, hidden Markov models (HMMs) were performed using the Blast method (E-value < e^−10^). Ultimately, we identified nine *BdTPS* and ten *BdTPP* genes, which were named *BdTPS1–BdTPS9* and *BdTPPA–BdTPPJ*, respectively (Table 1). The number of *TPS* genes was less than those in *Arabidopsis thaliana* (11), rice (*Oryza sativa*, 11), *Populus trichocarpa* (12), and soybean (13) [27] but more than in potato (8) [47] and drumstick tree (8) [45]. 

All of the basic information of the family members, including the locus, exon number, gene length, protein length, predicted isoelectric point (pI), and molecular weight (MW), is listed in Table 1. According to the results, BdTPSs/BdTPPs encode proteins with 820–985/351–421 amino acids (AA), predicted isoelectric points (pI) of 5.36–6.09 and 5.64–9.21, and molecular weights (MW) of 91.24–109.37 and 38.58–47.22 kDa, respectively. By predicting their subcellular localization, we found that only four *TPSs* (*BdTPS4*, *7*, *8*, *9*) were specifically located in cytoplasm, while the other *TPSs* were located in cytoplasm, inner membrane, periplasmic membrane, and/or outer membrane. In addition, three *TPP* genes (*BdTPPH*, *I*, *J*) were located in the cytoplasm, four *TPPs* (*BdTPPB*, *E*, *F*, *G*) were located in the periplasm, and three *TPPs* (*BdTPPA*, *C*, *D*) were located in both positions.

### 2.2. Gene Structure, Multiple Sequence Alignment, and Phylogenetic Analyses of BdTPSs/BdTPPs 

To investigate homologous relationships, an evolutionary tree was constructed using the BdTPS/BdTPP protein sequences with the MEGA 7.0 program [48]. According to the previous classification principles used for rice and *Arabidopsis TPS/TPP* genes [8,49], BdTPSs/BdTPPs can be divided into three subfamilies. Surprisingly, only the *BdTPS1* gene belonged to Class I (Figure 1a), which corresponded to the preceding results in rice, compared with four members (*AtTPS1–4*) in *Arabidopsis* and two in *Moringa oleifera* [21,50,51]. The other *BdTPSs* (*BdTPS2–9*) belonged to Class II, similar to *AtTPS5–11*. In addition, the Class III groups (*BdTPPA–BdTPPJ*) encoded a family of smaller proteins lacking the N-terminal TPS-like domain and with a conserved TPP catalytic site motif (Figure 1c).

In contrast to the similar CDS length (2463–2958) of the nine *TPS* genes, there are large differences in genomic length (2783–7172). However, all *TPP* genes have similar number of exons (6–10), gene lengths (1793–2835), and CDS lengths (1056–1266) (Table 1, Figure 2b). Moreover, the BdTPS1 protein sequences contain an N-terminal extension in comparison with other BdTPS proteins, which has been proven an autoinhibitory domain modulating TPS activity [51]. 

### 2.3. Protein Sequences and Motif Analysis in BdTPSs/TPPs

A homology analysis between Class I (BdTPS1) and eight other Class II TPS proteins (BdTPS2–9) indicated that the nucleotide identity between them ranged from 28.07% to 38.16%, while the nucleotide identity among the Class II proteins ranged from 40.12% to 81.87% (Figure 2a). Higher differences were observed in regions of the protein sequences outside of the TPS and TPP domains. The average identities of the peptide sequences of the TPS and TPP domains in BdTPS were about approximately 52%, while the sequence identity outside the domain was only 27% of the sequence identities (Figure 2b). Furthermore, the full protein sequences of the TPP family members had 40.23%–77.39% pairwise sequence identity (Figure 2c). Similarly, the outside domain in BdTPP showed a lower identity of 28% (Figure 2c), suggesting that the outside domain had a more variable sequence than the structural domain. 

In addition, to examine the differences in the TPP domains of BdTPS and BdTPP, we also calculated the protein pairwise sequence identities in these two families. The results revealed approximately 52% amino acid identity on average among BdTPSs and BdTPPs, while higher rates (65%) of pairwise sequence identities were observed in BdTPPs (Figure 2a,d). Extraordinarily, the TPP domain region between BdTPS1 and BdTPPs had only 27% pairwise sequence identity (Figure 2a). Furthermore, the protein sequences between the BdTPS and BdTPP groups had only 27% pairwise sequence identity (Figure 2d). 

Motif distribution analysis of all BdTPS/BdTPP proteins was performed by the MEME program. A total of ten conserved motifs of BdTPS/BdTPP are shown in Figure 1d. We summarized the three groups as follows. First, all BdTPS proteins of Class II contained nine motifs excluding motif 10. BdTPS1 possessed only seven motifs, lacking motifs 5 and 7 compared with Class II members. Moreover, there were two copies of motif 6 in BdTPS1. These findings support our previous results that BdTPS1 (Class I) has diverse sequences compared with other BdTPSs (Class II). Second, all of the BdTPPs contained five motifs, namely, motifs 5–10, which showed the conserved structure in all BdTPP proteins. Third, we found that the motifs 7 and motif 10 were specifically localized in the TPP domains of BdTPS and BdTPP, respectively. 

### 2.4. Chromosomal Locations and Evolution Analysis

To research the distribution of *BdTPS* and *BdTPP* genes in the genome, we showed their position on each chromosome based on the *B. distachyon* genomic database (Figure 3; Table 2). Both the *BdTPS* and *BdTPP* genes were dispersed on four chromosomes. However, we did not find *BdTPS* genes on chromosome 5 or *BdTPP* genes on chromosome 2. In detail, chromosome 1 contained five genes (*BdTPS8* and *BdTPPB*, *H*, *I*, *J*), chromosome 2 contained three genes (*BdTPS1*, *5*, *6*), chromosome 3 contained seven genes (*BdTPS2*, *3*, *4*, and *BdTPPA*, *C*, *E*, *G*), chromosome 4 contained three genes (*BdTPS3*, *9*, and *BdTPPF*) and chromosome 5 contained only one gene (*BdTPPD*).

Gene duplications on genomes could provide significant reference for gene evolution analysis. In the following research, were analyzed these two families. According to the standard for inferring a gene duplication event (>70% protein sequence identity and >75% sequence length identity) [52] and the results of phylogenetic analysis (Figure 1a), we predicted five clustered duplicated gene pairs (*BdTPS5–6*, *BdTPPC–D*, *BdTPPE–F*, *BdTPPG–I*, and *BdTPPH–J*) in the *B. distachyon* genome. We did not find any evidence of a tandem duplication phenomenon between these gene pairs according to the generalized principle that two genes should be separated by less than five genes in a 100 kb region [53]. According to the chromosome locations, two gene pairs (*BdTPS5–6*, *BdTPPH–J*) and three gene pairs (*BdTPPC–D*, *BdTPPE–F*, *BdTPPG–I*) may be generated by intrachromosomal duplication and interchromosomal duplication, respectively.

To understand the selective pressure acting on the trehalose biosynthesis enzyme homologue gene family, the *Ka/Ks* (non-synonymous/synonymous) values of these five duplicate gene pairs were calculated as shown in Table 2. The results indicated that the *Ka/Ks* values of the five pairs of duplicated genes were less than 1, indicating that they have evolved by purifying selection (*Ka/Ks* = 1, neutral selection, *Ka/Ks* < 1, purifying selection, and *Ka/Ks* > 1, positive selection [54]), which was similar to the results obtained in other plants [27,44,45].

Next, we calculated the divergence time for five pairs of duplicated genes. The results are listed in Table 2, which shows that two pairs of genes, *BdTPPE/F* and *BdTPPG/I*, were estimated to diverge at approximately 32.75 and 41.58 million year ago (MYA) and may represent two recently duplicated gene pairs. In contrast, the divergence times of one pair of genes (*BdTPPC/D*) surpassed the divergence time of grass species (approximately 56–73 MYA) [43]. In addition, two pairs of genes, *BdTPS5/6* and *BdTPPH/J*, existed at the time of the divergence time of grass species, prompting the concomitant formation of those two pairs of genes by whole-genome duplication.

### 2.5. Transcription Factor Regulatory Network and Stress-Related Cis-Elements in the Promoter Regions and Gene Interaction Expression Network Prediction of BdTPSs/BdTPPs

Many previous researches have demonstrated that the *TPS/TPP* genes confer biotic and abiotic stress tolerance on different plants in addition to developmental alterations. For example, the overexpression of *AtTPS1* significantly improved the drought resistance of *Arabidopsis* [55]. The overexpression of *OsTPS1* increased rice tolerance to abiotic stress [56]. *OsMAPK3* was found to phosphorylate *OsICE1*, inhibit its ubiquitination to activate the expression of the *OsTPP1* gene, and improve the chilling tolerance of rice [14]. In this study, we first predicted the stress-related *cis*-elements and co-expression network using different online tools.

First, we identified *cis*-elements of all *BdTPSs/BdTPPs* using an online tool, PlantCARE. Eight abiotic stress-response elements, abscisic acid (ABA)-responsive elements (ABREs), dehydration-responsive elements (DREs), low-temperature-responsive elements (LTREs), stress-response elements (TC-rich repeats LTRs, TGACG motifs), regulatory elements essential for anaerobic induction (AREs), and W-box were identified and are displayed in Figure 4. From the results, we found that almost all of the genes possessed more than two stress-related *cis*-elements. Remarkably, most of these family members possessed multiple ABREs, excluding *BdTPS1* and jasmonic acid methyl ester (MeJA), which respond to ABA, drought or salt signals, and drought, respectively [57,58,59,60]. Furthermore, many other elements were distributed among different family members, such as MBS elements involving drought stress (*TPS2* and *TPPB*, *C*, *E*, *F*, *I*, *J*) [61]; DREs concerning dehydration, low-temperature and salt stresses (*TPPB*, *C*, *E*, *F*, *J*); TC-rich elements (*TPPG* and *J*); LTREs, involved in low-temperature responses (*TPS1*, *2*, *4* and *TPPA*, *B*, *C*, *F*, *H*, *J*); AREs essential for anaerobic induction (*TPS4*, *6*, *7*, *8* and *TPPA*, *B*, *D*, *E*, *F*, *G*, *H*, *I*); and W-boxes, binding sites for WRKY transcription factors in stress responses (*TPS2*, *3*, *4*, *5*, *9* and *TPPA*, *B*, *G*, *H*, *I*, *J*). Regrettably, we did not find the HSE element that was involved in the heat-resistance response. In addition to these stress-response elements, plentiful light-response elements were identified, such as the ACE, AE-box, G-box, I-box, and TCT motifs.

Furthermore, to identify the potential transcriptional regulatory network of the *BdTPS* and *BdTPP* genes, sequences 600 bp upstream of the genes were analyzed. The results indicated hundreds of the transcription factors involved in the regulation of 19 family members, mainly including MYB, ERF, and bZIP and many other families (Figure 5). Among these family members, many of the transcription factors were related to the response to abiotic stress or phytohormones (Appendix A), corresponding to the result of the *cis*-element analysis.

In addition, to obtain information regarding the co-expression relationships between *BdTPS/BdTPP* and other genes, we constructed a co-expression network using the STRING and PlaNet co-expression databases [62]. First, we used the online program STRING to predict the co-expression networks in 19 family members. The results showed strong ties between these two family members (Figure 6). Second, we also predicted the gene interaction connection on (take *BdTPS1* for example) single genes based on PlaNet, which showed several directly and indirectly interacting genes (Figure 7). The GO annotations of directly interacting genes involved in diverse biological functions and are presented in Appendix A. Most BdTPS/BdTPP proteins interact with many of these other family members. In particular, some genes may work in response to chilling or other stresses (Appendix A). However, there were no co-expression network data on any of the proteins, including *BdTPS3*, *8*, and *BdTPPA*, *J.*

### 2.6. Expression Patterns of BdTPSs–BdTPPs in Different Tissues and Stress Conditions

In this study, quantitative real-time reverse transcription (qRT)-PCR was used to examine the expression patterns of the genes *BdTPSs-BdTPPs* in different tissues, including roots, stems, leaves, and spikes (Figure 8 and Appendix A). All of the *BdTPS* and *BdTPP* genes were expressed in all tissues, although majority of the *BdTPP* genes were expressed at lower levels. In addition, *BdTPS3*, *BdTPPH*, and *BdTPPI* were expressed in four tissues at very high levels. Some genes, such as *BdTPS2*, *BdTPS9*, *BdTPPC*, and *BdTPPG*, were strongly expressed in roots; *BdTPS2* was strongly expressed in stems. Some genes, such as *BdTPS5* and *BdTPPE* in the spikes, *BdTPPB* and *BdTPPF* in the stems and spikes, and *BdTPPD* and *BdTPPJ* in the leaves and spikes, were expressed at very low levels.

To further investigate the functions of the *BdTPSs* and *BdTPPs* in response to abiotic stresses, we determined the expression patterns of family members under three different stress conditions: 4 °C, drought and salinity stress (Figure 9, and Appendix A). As shown in Figure 9 and Figure 10, many of the *BdTPS* genes and some of the *BdTPP* genes may be related to various abiotic stress tolerances. Under cold treatment, four genes (*BdTPS5*, 9 and *BdTPPC*, *E)* were upregulated, one gene (*BdTPS3*) was stabilized, and the other genes were downregulated. After exposure to 200 mM NaCl, eight genes (*BdTPS3*, *4*, *5*, *6*, *7*, *9*, and *BdTPPA*, *C*) were upregulated, and the other genes were downregulated. After drought stress, seven genes (*BdTPS3*, *5*, *6*, *7*, *8*, *9*, and *BdTPPC*, *E*) were upregulated at different times. Strikingly, the expression levels of specific genes such as *BdTPS5*, *BdTPS6*, *BdTPS9*, and *BdTPPC* increased under cold, salinity, and dehydration stress, which was similar to the behavior of rice *OsTPS1*.

ABA, a crucial phytohormone induced by biotic or abiotic stress, plays important roles in plant tolerance to abiotic stresses [63]. In this paper, we also examined the gene expression patterns after treatment with 100 µM ABA solution. The results showed that only *BdTPPE* had a higher expression level at all three time points; some genes, such as *BdTPS5*, *6*, *7*, *9*, and *BdTPPA*, *F*, were upregulated after 6 h of exposure, while most other genes were downregulated at all three times points. 

## 3. Discussion

As many species were sequenced, an increasing number of TPS and TPP family members have been identified. The TPS family has been reported in several higher plants, such as *A. thaliana*, *O. sativa*, *Nelumbo nucifera*, *Solanum tuberosum*, *M. oleifera*, *Gossypium*, and *Triticum aestivum* [8,26,44,45,46,47,64]. However, few studies have systematically reported the TPP family, even though this family, similar to the TPS family, is ubiquitous throughout the plant kingdom. Accumulating evidence has revealed that some TPS/TPP family members are related to responses to abiotic stresses and potentially function in improving stress tolerance and increasing crop yields [14,41]. As *B. distachyon* is a model plant of Gramineae, it is very important to understand the function and evolution of its TPS and TPP families. 

In this research, we identified nine *BdTPS* and 10 *BdTPP* genes in *B. distachyon*. All of the *BdTPS* genes contained Glyco_transf_20 (TPS) and Trehalose_PPase (TPP) domains, and all of the *BdTPP* members included only Trehalose_PPase domains. The trehalose biosynthesis enzyme homologue gene families have been classified as three groups according to their protein sequences and phylogenetic relationships. Unlike the four Class I genes distributed among the *A. thaliana* genome, only one Class I member has been identified in *B. distachyon* with an N-terminal extension, which has been demonstrated to be an inhibitory domain regulating TPS activity [21,50]. This phenomenon was also consistent with previous studies on *A. thaliana* (*AtTPS1*), *O. sativa* (*OsTPS1*), *Selaginella lepidophylla* (*SlTPS1*), and *M. oleifera* (*MoTPS1*) [8,41,45] and may affect the content of trehalose by regulating the *TPS1* gene in plants. We do not have enough evidence to explain why there was only one member of Class I *TPS* in *Brachypodium* and rice. We hypothesized that the Class I *BdTPSs* experienced slower duplication speeds than the Class II *BdTPSs* over the course of genome evolution. Some traits differed between the two TPS groups, such as gene structure, gene length, amino acid identity, and motif structure. Alignment analysis showed that the Class I group gene, *BdTPS1*, included 17 exons, in contrast to three exons in other *BdTPS* genes (Table 1, Figure 1b). Previous theories suggested that the rate of intron gain was slower than the rate of intron loss after duplication [44,65]. Therefore, we speculated that *BdTPS1* might be the ancestor of other *BdTPS* genes. By motif analysis, we found that the motif 4–9 region was conserved in Class II BdTPSs instead of in BdTPS1 proteins, suggesting that these differences between the two classes may be involved in various functions. In contrast, *BdTPPs* have similar characteristics, suggesting they might have similar gene functions.

The sequence alignments indicated that both domains were conserved among *BdTPS* genes, suggesting that they were mainly formed before the differentiation of these genes. This finding agreed with preceding observations in other plants [27,45]. However, we also noticed that the TPP domains have higher sequence differences between BdTPSs and BdTPPs (Figure 2d). Additionally, by comparison of the motif sequences of the TPP domains in these two families, we found that motif 7 was specific to BdTPS whereas motif 10 was specific to BdTPP. In *Arabidopsis*, AtTPSs contain a TPP-like domain with no TPP catalytic function, while all of the AtTPPs contain a conserved TPP catalytic site domain [49]. It seems that the TPP domain was either functionally differentiated between the two families or catalytically inactivated in the TPS family. Interestingly, the outside domains of the protein sequences for both BdTPS and BdTPP also showed lower identity than the TPP and/or TPS domains suggesting that these outside domain regions may contribute to the functional differentiation.

Further analysis showed that four pairs of *BdTPP* genes were formed by duplication events, suggesting that gene duplication might play a vital role in the expansion of the *BdTPP* genes. Similarly, four pairs of *AtTPP* genes expanded exclusively upon genome duplication in *Arabidopsis*, except for *AtTPPA* and *AtTPPD* [24], implying similar expansion patterns of TPP in both monocot and dicot plants. In contrast, only one pair of *BdTPS* genes was relevant to duplication events, implying that most of the *TPS* genes were formed a very long time ago by duplication events. According to the results of the estimated divergence time analysis, we speculated that two pairs of genes (*BdTPPE/F* and *BdTPPG/I*) might represent two newly duplicated gene pairs after a whole-genome duplication event; two pairs of genes (*BdTPPH/J* and *BdTPS5/6*) might have duplicated together with the whole-genome duplication event; and one pair of genes (*BdTPPC/D*) might have existed before *Brachypodium* diverged from other grass species. This small-scale gene duplication phenomenon in *BdTPP* was different from whole-genome duplications in *AtTPPs*. The evolutionary driving force acting on the *BdTPS* and *BdTPP* genes was purifying selection, which was similar to what has been observed in other plants [27,44,45].

Many previous studies have found that *TPS/TPP* genes provide abiotic stress tolerance in different plant species, along with developmental alterations. For example, overexpressed *AtTPS1* has significantly improved the drought resistance of *Arabidopsis* [55]. Overexpressed *OsTPS1* could increase rice tolerance to abiotic stress [56]. The activation of *OsTPP1* expression can improve chilling tolerance in rice [14]. In this study, we first predicted stress-related *cis*-elements and co-expression networks using different online tools. All of the results provided a similar conclusion: Namely, the *BdTPS/BdTPP* gene might be involved in regulating stress tolerance together with other genes. In brief, the co-expression network prediction of *BdTPS/BdTPP* genes may provide a basic reference for better understanding the signal path in *B. distachyon*.

ABA, a crucial phytohormone induced by biotic or abiotic stress, plays important roles in plant tolerance to stresses [63]. Our results showed that only *BdTPPE* was heavily induced after exposure to ABA, strongly supporting that *BdTPPE* may be related to the ABA signaling pathway. Other genes (*BdTPS5*, *6*, *7*, and *BdTPPA*, *F*) may also be related to the ABA signaling pathway because these genes were upregulated with increasing treatment time and reached their highest expression levels in 6 h.

Furthermore, we examined the expression patterns of *BdTPS* and *BdTPP* genes under cold, salinity, and dehydration stress. As shown in Figure 9 and Figure 10, many of the *BdTPS* genes and some *BdTPP* genes might connect with various aspects of abiotic stress tolerance. Furthermore, most of these genes might be involved in two different stress tolerances. For instance, *BdTPS7* might be involved in salt and drought resistance, while *BdTPPE* might participate in cold and drought tolerance. Extraordinary, the expression levels of some specific genes, such as *BdTPS5*, *BdTPS6*, *BdTPS9*, and *BdTPPC*, increased under chilling, salinity, and dehydration stress, similar to what has been shown in rice *OsTPS1* [56]. We speculated that these overexpressed genes might improve stress tolerance for *B. distachyon*. In addition, we noticed that some family members related to salinity or drought resistance, such as *BdTPS5*, *6*, *7*, *9*, and *BdTPPA*, *E*, also responded to ABA, and ABA-responsive elements (ABRE) have also been found on the promoter regions of these genes (Figure 4), suggesting that these family members might be connected with response to dehydration or salinity by participating in ABA signaling pathways. In contrast, we found that *BdTPS2* and five *TPP* genes (*BdTPPB*, *G*, *H*, *I*, *J*) might not be related to these three different abiotic stresses because the expression levels of these genes were downregulated.

## 4. Conclusions

In this study, we identified nine *BdTPS* and ten *BdTPP* genes on the *B. distachyon* genome and analyzed their evolution by evaluating exon/intron structures, protein motifs, phylogenetic relationships, and *Ka/Ks* values. These family members were classified as three classes based on an evolutionary tree. Gene structure analysis showed that *BdTPS1*, which is the only Class I gene, might be the ancestral gene. All of the *TPS* and *TPP* genes have evolved mainly by purifying selection. In addition, only one pair of *BdTPS* genes and four pairs of *BdTPP* genes expanded in *B. distachyon* by chromosomal duplication. The tissue specificity profiles of the *BdTPS* and *BdTPP* genes in four different tissues suggested that most of these genes were expressed in all tissues at various levels. The expression patterns under various stress conditions showed that most of the *TPS* genes and approximately half of the *TPP* genes may be involved in stress resistance. In particular, *BdTPS5*, *BdTPS6*, *BdTPS9*, and *BdTPPC* might be broad-spectrum abiotic stress resistance genes under abiotic stress conditions. This study provides a foundation for future study on the biological function of *BdTPSs* and *BdTPPs.*

## 5. Materials and Methods

### 5.1. Gene Identification

*BdTPS* and *BdTPP* gene identification was based on the hidden Markov model (HMM) with an E-value < 1e^−10^ [66]. The TPS domain (glycosyltransferase family 20 (Glyco_transf_20); PF00982) and TPP domain (trehalose-phosphatase (Trehalose_PPase); PF02358) were obtained from the Pfam database (http://pfam.xfam.org/). In addition, the protein sequences of the *Arabidopsis* and rice *TPS/TPP* were used as queries for BLASTN searches in the Phytozome database (https://phytozome.jgi.doe.gov/pz/portal.html#). Finally, all candidate genes were examined for the TPS and TPP domains (TPPs only) in the Pfam and SMART databases (http://smart.embl-heidelberg.de/). Both genome data and TPS/TPP protein sequences were obtained from the Phytozome database.

### 5.2. Gene Structure, Protein Properties, and Conserved Motif Analysis

Motifs in the BdTPS and BdTPP protein sequences were predicted using the online program MEME (http://meme-suite.org) with the following parameters: The maximum number of motifs was set to 10, any number of repetitions was chosen for the site distribution, and the optimum motif width was set to ≥6 and ≤50. Subcellular localizations were predicted with ‘CELLO v.2.5′ (http://cello.life.nctu.edu.tw/). The Compute pI/MW tool (https://web.expasy.org/compute_pi/) was used to predict the isoelectric point (pI) and molecular weight (MW).

### 5.3. Prediction of Cis-Elements, Transcription Factor Regulatory Network, and Gene Interaction Network

The *cis*-elements were identified by downloading the 1500 bp upstream sequences [67] of the *BdTPS/BdTPP* genes from the Phytozome database and then submitting the sequences to PlantCARE (http://bioinformatics.psb.ugent.be/webtools/plantcare/html/) to identify seven stress-related regulatory elements: ABA-responsive elements (ABREs) [68]; dehydration-responsive elements (DREs) [69]; low-temperature-responsive elements (LTREs) [69]; regulatory elements essential for anaerobic induction (AREs) [70]; TC-rich repeats, involved in defense and stress response [71]; MYB-binding sites, involved in drought inducibility (MBS) [71]; jasmonic acid methyl ester (MeJA) responsiveness (CGTCA motif and TGACG motif) to drought stress [72]; and W-boxes, the binding site of *WRKY* transcription factors in defense responses [68]. The co-expression network was constructed by the online tool STRING (http://string-db.org); the single gene expression network was predicted using the PlaNet (http://www.gene2function.de/) [62,73]. The transcription factors were then predicted using the PTRM database (http://plantregmap.cbi.pku.edu.cn/regulation_prediction.php) with 600 bp sequences upstream from the start codon of the *TPS/TPP* genes [73].

### 5.4. Sequence Alignment and Phylogenetic and Evolution Analyses

For the phylogenetic analyses, full-length BdTPS and BdTPP protein sequences from *B. distachyon* were aligned using ClustalX2.11 [45], and the alignment result was used to construct the phylogenetic tree with MEGA 7 using neighbor joining methods [48]. The exon/intron structures of *BdTPSs* and *BdTPPs* were analyzed with Gene Structure Display Server 2.0 (http://gsds.cbi.pku.edu.cn/). 

The chromosomal distribution was analyzed with the TBtools software package (http://www.tbtools.com/) [74]. The values of synonymous substitution (*Ks*) and non-synonymous substitution (*Ka*) were calculated using the web server with the Nei–Gojobori method (https://github.com/tanghaibao/bio pipeline/tree/master/synonymous_calculation). The divergence time (T) was computed according to the formula T = *Ks*/2λ based on the *Ks* rate of λ (λ = 6.5 × 10^−9^ for *B. distachyon*) substitutions per site per year [75,76].

### 5.5. Plant Growth, Stress Treatment, and qRT-PCR Analysis

The diploid inbred line Bd21 was used for the gene expression level-related experiments. First, seeds were sterilized and germinated in water-soaked filter paper for three days in darkness at 25 °C [77]. Second, seedlings were removed to a cultivating basin with Hoagland solution (5 mM Ca (NO_3_)_2_, 5 mM KNO_3_, 2 mM MgSO_4_, 1 mM KH_2_PO_4_, 0.4 μM CuSO_4_, 50 μM FeNa_2_(EDTA)_2_, 50 μM H_3_BO_3_, 10 μM MnC_12_, 0.02 μM, (NH_4_)_6_MoO_24_, and 0.8 μM ZnSO_4_) in the greenhouse under a 16/8 h (light/dark) photoperiod at 22 °C [77]. Three different plant tissues (roots, stems, and leaves) were collected after three weeks; spikes were collected from two-month-old Bd21. During the experimental period, the three-week-old seedlings were exposed to low temperature (4.0 °C), salt (200 mM NaCl), simulated drought (20% (w/v) PEG 6000, Coolaber, Beijing, China), and ABA solution (100 μM ABA, Solarbio, Beijing, China). Whole plants were collected, 1, 3, and 6 h after exposure to the abiotic stress in three biological replicates. All samples were immediately frozen in liquid nitrogen and stored at −80 °C until further processing for RNA extraction. Total RNA was extracted using a HiPure Plant RNA Mini Kit (Magen, Guangzhou, China) and cDNA synthesis was performed using StarScript II (GeneStar, Beijing, China). Specific primer pairs and reference genes (*SamDC*) were designed by qPrimerDB and the ICG database, respectively [78,79] and are listed in Appendix A. Transcript levels were quantified using a CFX96 Real-Time PCR Detection System (Bio-Rad, Hercules, USA) under the following conditions: 95 °C for 2 min and 40 cycles of 95 °C for 15 s and 60 °C for 20 s [80]. Four reactions and three independent biological replicates were performed. The relative quantitative analysis of the genes was based on the 2^−ΔΔCT^ method [81]. The expression value of the *BdTPS 1* gene in the root is defined as 1. The heat map of 29 genes was constructed using TBtools software. Statistical analyses were performed using a *t*-test, and p < 0.05 was considered a significant difference. 

## Figures and Tables

**Figure 1 plants-08-00362-f001:**
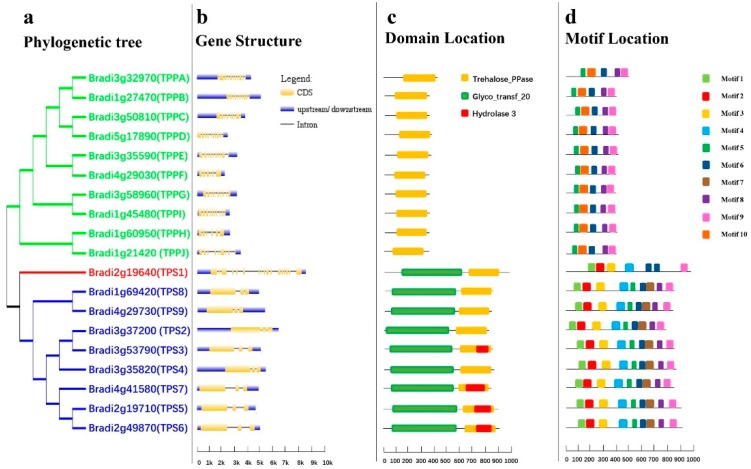
Phylogenetic relationships and gene structures, domains, and motifs of the *BdTPS* and *BdTPP* family. (**a**) Phylogenetic analysis, Class I, II, and III genes are shown in red, blue, and green, respectively. (**b**) Gene structure, introns, exons, and upstream/downstream regions are shown as straight lines and yellow and blue bars, respectively. (**c**) The conserved trehalose-6-phosphate synthase (TPS) domain (Glyco_transf_20), trehalose-6-phosphate phosphatase (TPP) domain (Trehalose_PPase), and haloacid dehalogenase-like hydrolase domain containing 3 (Hydrolase 3) are shown by yellow, green, and red, respectively. (**d**) Motif analysis, all motifs were identified by MEME tools, as shown in different bars.

**Figure 2 plants-08-00362-f002:**
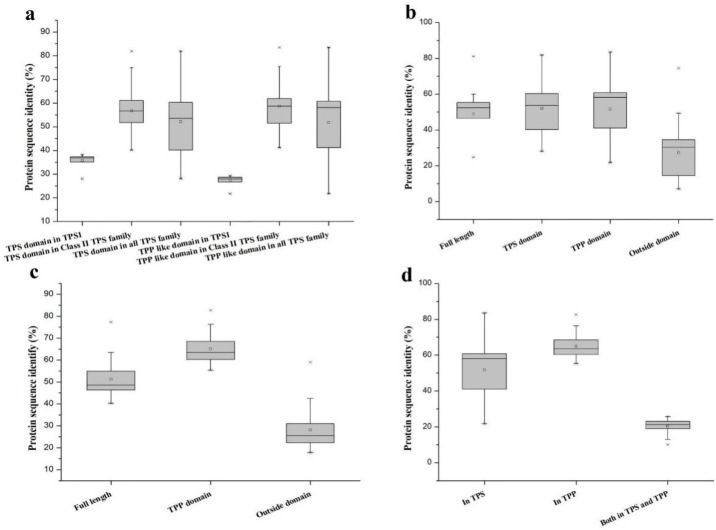
Pairwise sequence identities for different regions of the *B. distachyon* TPS and TPP proteins. (**a**) Pairwise sequence identities of the TPS and TPP domains in BdTPS. (**b**) Comparison of the TPS domain, TPP domain, protein sequence, and sequence outside the domain in BdTPS. (**c**) Comparison of the TPP domain, full-length protein sequence, and sequence outside the domain in BdTPP. (**d**) Comparison of the TPP domains in BdTPS and BdTPP.

**Figure 3 plants-08-00362-f003:**
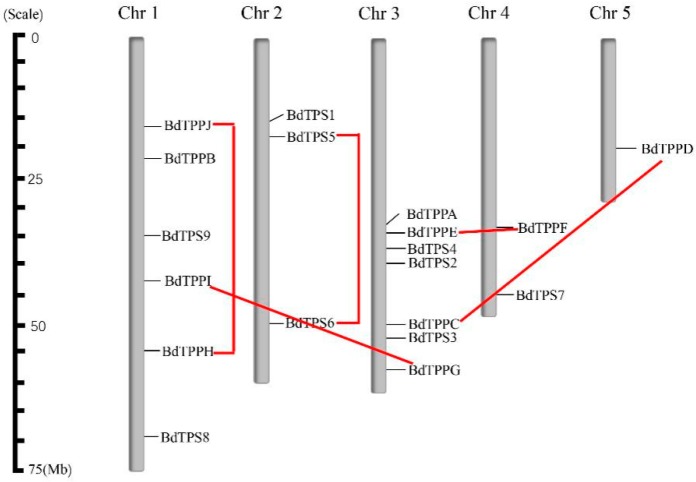
Chromosomal location and gene duplication of *BdTPSs/BdTPPs.* The chromosome numbers are indicated above, and red lines represent the duplication gene pairs.

**Figure 4 plants-08-00362-f004:**
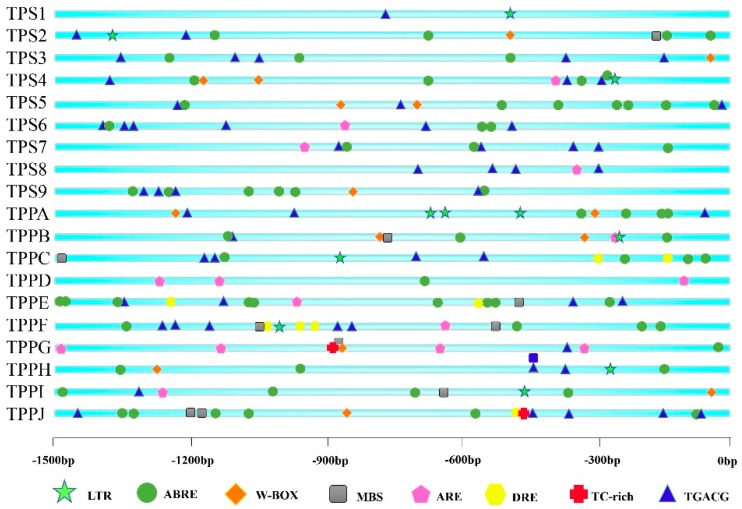
Predicted *cis*-elements related to stress resistance in the *BdTPS/BdTPP* promoters. Promoter sequences (−1500 bp) of 19 genes were analyzed using the PlantCARE database. The different colors and shapes of markers represent various *cis*-elements.

**Figure 5 plants-08-00362-f005:**
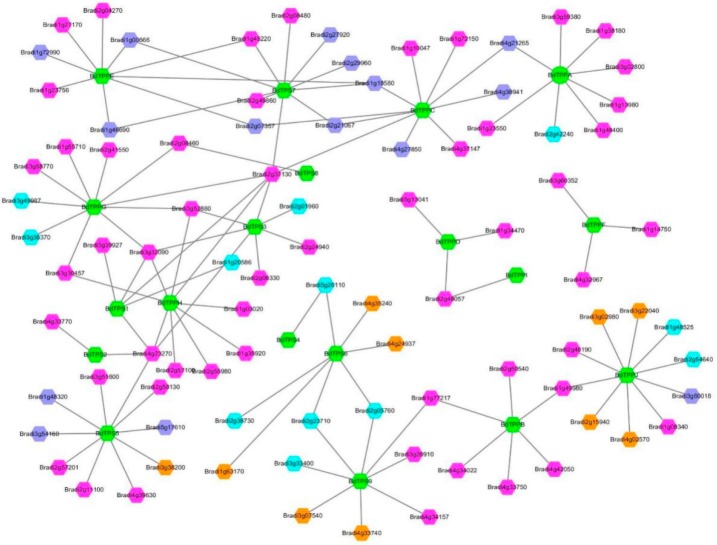
Prediction of the transcriptional regulatory network. The green hexagon represents *BdTPS/BdTPP* genes. Different colors represent different transcription factor families, with brilliant blue, dark blue, orange and purple hexagons indicating MYB, ERF, bZIP and 21 other transcription factor families, including B3, bHLH, C2H2, MIKC_MADS, TCP, LBD, NAC, BES1, EIL, Dof, E2F/DP, G2-like, Nin-like, CAMTA, CPP, Trihelix, ARF, C3H, GATA, TALE, and BBR-BPC family members.

**Figure 6 plants-08-00362-f006:**
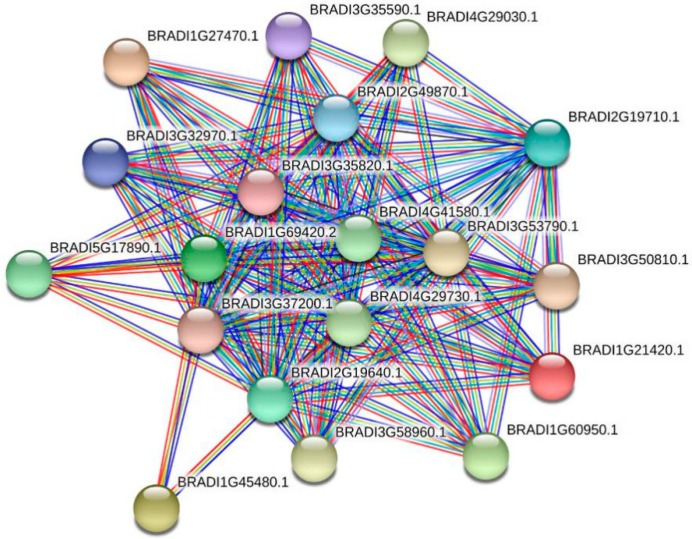
The interaction network between members of the family. Protein interaction network prediction showed that most BdTPS and BdTPP proteins interact with more than one other family member. Line colors represent the various types of evidence used to predict the protein interaction network.

**Figure 7 plants-08-00362-f007:**
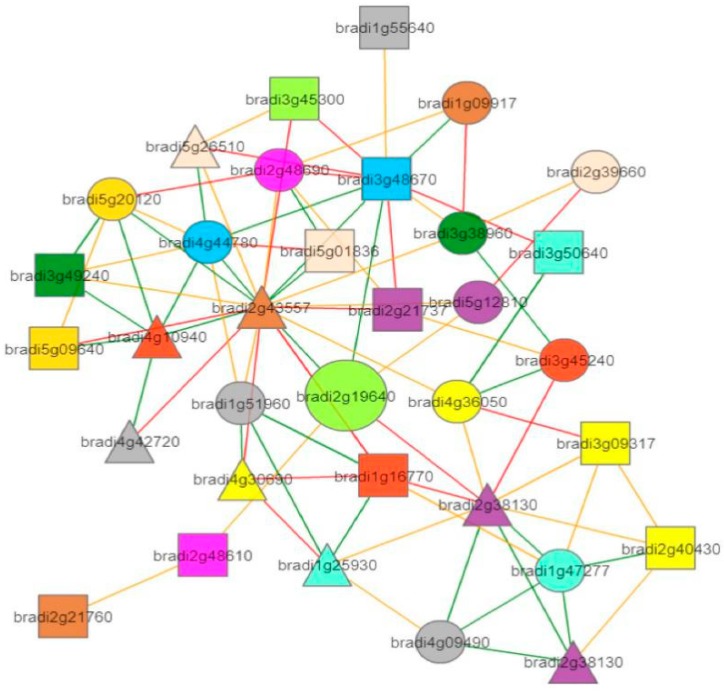
Co-expression neighborhood of PsaN (take *BdTPS1* for example, large light green circle). The straight lines of different colors show how strongly two genes are co-expressed, with green, yellow, and red edges indicating strong, medium, and weak co-expression, respectively.

**Figure 8 plants-08-00362-f008:**
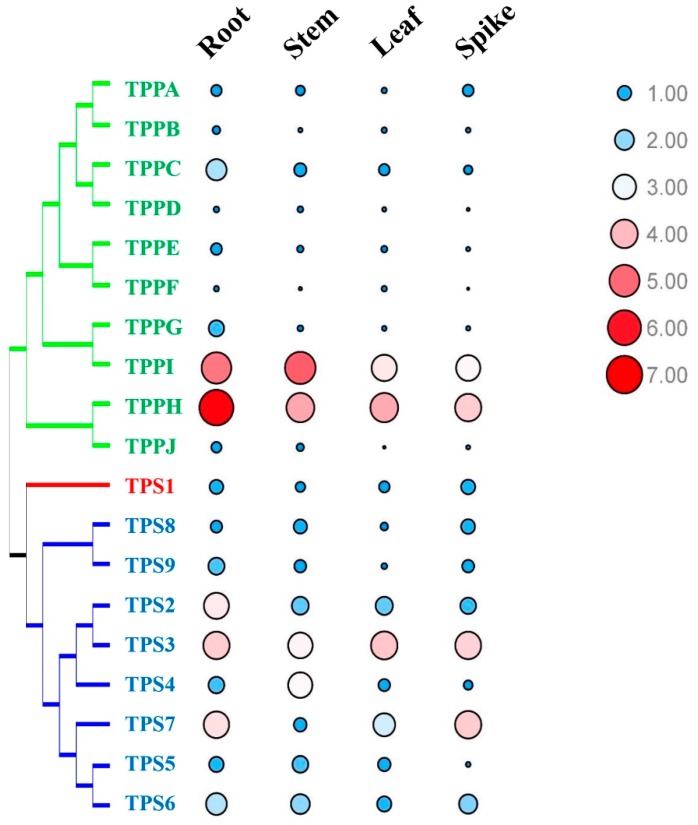
Expression profiles of *BdTPS* and *BdTPP* genes in four tissues. Blue color and small circle/red color and large circle that represent low/high expression, respectively.

**Figure 9 plants-08-00362-f009:**
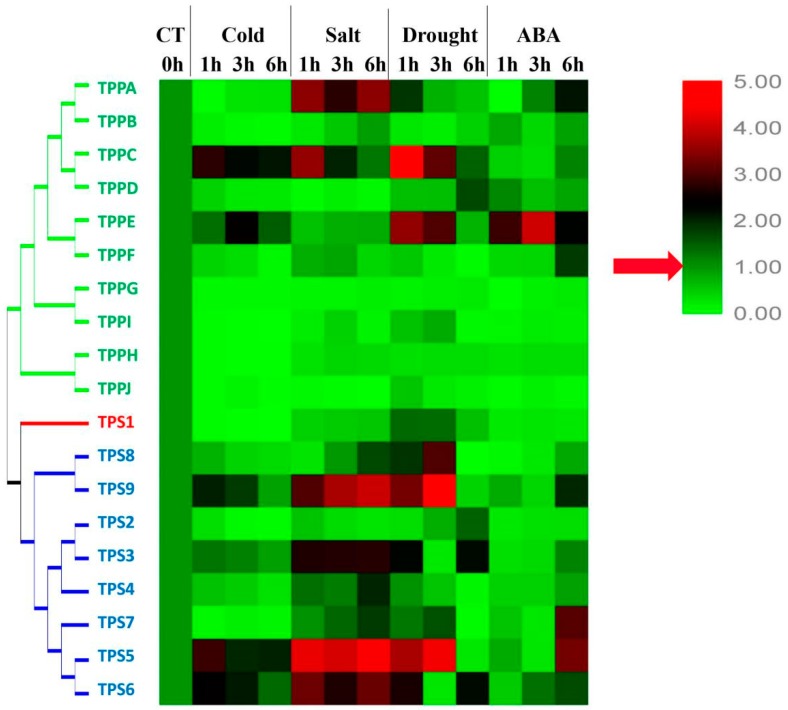
Expression patterns of *BdTPS* and *BdTPP* genes under three different abiotic stresses and abscisic acid (ABA) treatment in the whole plants. The gradient ramp represents different relative expression levels. The green indicates a low expression level and the red indicates a high expression level.

**Figure 10 plants-08-00362-f010:**
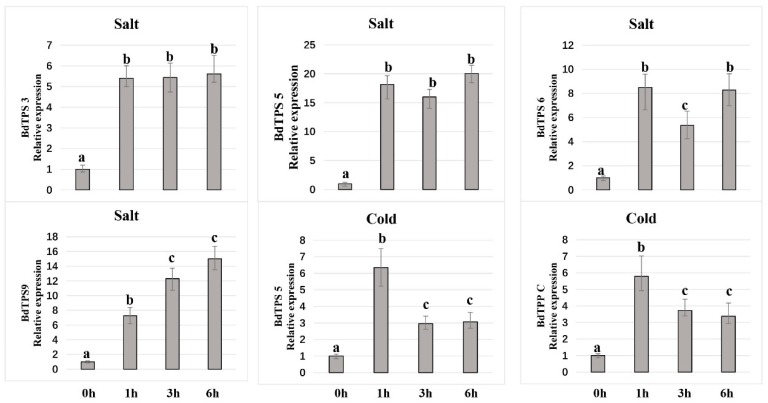
Relative expression level of some *BdTPS/BdTPP* genes under salt and cold stress conditions at 1, 3, and 6 h after exposure to the abiotic stress. Different lowercase letters (a, b, c) above the bars indicate significantly different values (*p* < 0.05, *t*-test).

**Table 1 plants-08-00362-t001:** Basic characteristics of *TPS* and *TPP* genes in *Brachypodium distachyon.*

Name	Gene ID	Locus	Exon Number	Gene Length	CDS (bp)	No. of AA	pI	MW (kDa)	PSL
*BdTPS1*	Bradi2g19640	17272345..17280887	17	7172	2958	985	6.09	109.37	C,P,OM
*BdTPS2*	Bradi3g37200	39265879..39272239	3	3409	2463	820	6.04	91.24	C,IM
*BdTPS3*	Bradi3g53790	54157803..54162799	3	3375	2556	851	6.47	94.76	C,P,IM
*BdTPS4*	Bradi3g35820	37837473..37842899	3	2783	2589	862	6.06	97.71	C
*BdTPS5*	Bradi2g19710	17341436..17346150	3	3783	2709	902	5.36	100.85	C,OM
*BdTPS6*	Bradi2g49870	49645444..49650378	3	4238	2730	909	5.47	101.74	C,OM
*BdTPS7*	Bradi4g41580	45574020..45578868	3	3755	2634	877	5.82	98.58	C
*BdTPS8*	Bradi1g69420	68065706..68070557	3	3067	2604	867	5.99	97.75	C
*BdTPS9*	Bradi4g29730	35237459..35242687	3	2823	2610	869	5.99	97.75	C
*BdTPPA*	Bradi3g32970	35087863..35092068	11	2203	1266	421	6.24	47.22	P,C
*BdTPPB*	Bradi1g27470	22586832..22591975	10	1874	1056	351	6.16	39.19	C
*BdTPPC*	Bradi3g50810	51669711..51673415	9	1900	1119	372	5.68	41.06	P,C
*BdTPPD*	Bradi5g17890	21178294..21180648	10	1999	1077	358	8.52	39.93	P,C
*BdTPPE*	Bradi3g35590	37622570..37625654	10	2300	1098	365	9.21	40.52	P
*BdTPPF*	Bradi4g29030	34348411..34350663	6	1793	1158	385	8.98	41.79	P
*BdTPPG*	Bradi3g58960	58022082..58025342	10	2243	1101	366	8.59	39.92	P
*BdTPPH*	Bradi1g60950	60533219..60535884	6	2071	1080	359	6.22	39.08	C
*BdTPPI*	Bradi1g45480	43862356..43865024	10	2266	1101	366	6.27	40.05	C
*BdTPPJ*	Bradi1g21420	17249655..17253046	9	2835	1062	353	5.64	38.58	C

Note: CDS, protein length in bp; pI, isoelectric point; PSL, predicted subcellular localization; No. of AA, number of amino acids; C, cytoplasmic; P, periplasmic; OM, outer membrane, IM, inner membrane.

**Table 2 plants-08-00362-t002:** The duplicated *BdTPS/BdTPP* genes and *Ka/Ks* analysis.

Gene Pairs	Identity (%)	*Ka*	*Ks*	*Ka/Ks*	MYA
*BdTPS5–BDTPS6*	81.10	0.1108	0.8821	0.1256	67.85
*BdTPPC–BdTPPD*	63.44	0.2185	1.0765	0.2030	82.80
*BdTPPE–BdTPPF*	74.16	0.1278	0.4258	0.3000	32.75
*BdTPPG–BdTPPI*	63.64	0.2354	0.5406	0.4354	41.58
*BdTPPH–BdTPPJ*	63.66	0.2619	0.8726	0.3001	67.12

Note: Identity, protein sequences similarity; *Ka*, non-synonymous; *Ks*, synonymous; MYA, million years ago.

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
