# Peer review of "Genome-Wide Identification, Evolution, and Expression Analysis of TPS and TPP Gene Families in Brachypodium distachyon"

_plants, 2019, doi:10.3390/plants8100362_

Round 1

Reviewer 1 Report

Wang et al. identified trehalose-6-phosphate synthase and trehalose-6-phosphate 3 phosphatase gene families in the genomes of Brachypodium distachyon L. and examined their gene structures, chromosomal localization, evolutionary history, and regulatory elements. They also examined their expression patterns in several tissues and the responses to various stresses. Because the role of trehalose in stress response has been studied in other organisms, the genes involved in trehalose synthesis in higher plants is an important subject in order to understand the mechanisms of their adaptation to natural stresses.

The weakness is the evolution part. For instance, authors discuss that ”most of the TPS genes were formed independently instead of via gene duplication” on page 28. It is unlikely that gene family members are formed independently. They must have formed a very long time ago by duplication event and are sufficiently diverged to become lower than the 75% of identity between each gene. Also, authors should know more about how new genes are born. There are other important mechanisms for creating new genes in the genome, and such mechanisms may better explain why BdTPS1 gene has 17 exons whereas other BdTPS genes have only 3 exons.

Results from gene expression experiments are summarized in the heatmap in Fig. 7. Although the presentation looks comprehensive, there is no information on the significance of differences. The studies are done in triplicates so it would be better to show that the differences in a subset of cases, for instance, the salt treatment on TPS9 and TPPC, drought treatment on TPPC and TPPE, are significant. One way to do so is showing the fold induction by histograms and test the statistical significance, as most of the people do.

There are numerous typos and ambiguous descriptions. Authors should have looked at the manuscripts very carefully before submission. Listed here are only minor examples. There must be many others.

families “based on” should be “in.” I would say all the gene families are formed by gene duplications. I would say, “Gene expression analysis shows the tissue-specific pattern of TPS and TPP genes,”---- Yeast is not a prokaryote. Do you mean “higher plants”? I would say, “For the phylogenetic analyses, 150 full-length BdTPS and BdTPP protein sequences from (insert species names here) were aligned using the ClustalX2.11 program.

219-222. There is a redundant description.

239-240. Another redundant description.

found, not founded. is extrons a correct term? You don’t need “it was.” Identities, not identifies. That and? noted, you need to use something else. we searched for segmental duplication blocks ----. You need to explain how segmental duplications are defined in this search engine. because, to call tandem duplication, there should be less than five genes between these two genes. You can still call them intrachromosomal duplication or local duplication events. I think the “principle” authors employed is meaningless. This is interchromosomal duplication. You don’t need to regret. indicated or identified. Which one? low levels extra space found, not founded but not in class 1 gene

481-483. Unclear. Consider rewriting this part.

488-490. by comparing the motif sequences ---, we found that motif 7 was specific to BdTPS whereas motif 10 was specific to BdTPP.

504-505. This argument is very unlikely.

chilling tolerance in rice. reconsider the writing of this sentence. more than one?

Author Response

Dear editior and Reviewers,

Thank you very much. This comments are very valuable and very helpful for revising and improving our paper, as well as the important guiding significance to our researches. We have studied comments carefully and have made correction which we hope meet with approval. The main corrections in the paper and the responds to the reviewer’s comments are as following:

Wang et al. identified trehalose-6-phosphate synthase and trehalose-6-phosphate 3 phosphatase gene families in the genomes of Brachypodium distachyon L. and examined their gene structures, chromosomal localization, evolutionary history, and regulatory elements. They also examined their expression patterns in several tissues and the responses to various stresses. Because the role of trehalose in stress response has been studied in other organisms, the genes involved in trehalose synthesis in higher plants is an important subject in order to understand the mechanisms of their adaptation to natural stresses.

Point 1:The weakness is the evolution part. For instance, authors discuss that ”most of the TPS genes were formed independently instead of via gene duplication” on page 28. It is unlikely that gene family members are formed independently. They must have formed a very long time ago by duplication event and are sufficiently diverged to become lower than the 75% of identity between each gene. Also, authors should know more about how new genes are born. There are other important mechanisms for creating new genes in the genome, and such mechanisms may better explain why BdTPS1 gene has 17 exons whereas other BdTPS genes have only 3 exons.

Response 1: Thank you very much. I also consider this conclusion to be less than rigorous. I agree with your comment that ‘They must have formed a very long time ago by duplication event’, so I revised this error. Please see the manuscript.

According to previous results, the Class I gene in other plants, such as in rice, Arabidopsis, Populus, drumstick tree, also contains many exons. I think this phenomenon is common. Many studies have shown that only Class I TPS proteins had TPS activity. The contrast in gene structures and functions between class I and class II TPS members suggested their evolutionary divergence. I agree the point that the rate of intron gain is slower than that of intron loss after duplication (10.3389/fpls.2016.01445; 10.1016/j.gene.2010.06.008; 10.1038/srep22783). In addition, loss of AtTPS1 produces a lethal phenotype, indicating that this Class I gene plays an essential role in plant development. This evidence may support TPS1 as the ancestor of other TPS genes.

Point 2: Results from gene expression experiments are summarized in the heatmap in Fig. 7. Although the presentation looks comprehensive, there is no information on the significance of differences. The studies are done in triplicates so it would be better to show that the differences in a subset of cases, for instance, the salt treatment on TPS9 and TPPC, drought treatment on TPPC and TPPE, are significant. One way to do so is showing the fold induction by histograms and test the statistical significance, as most of the people do.

Response 2: Thank you for this comment. I added the Figure 10 to show the case according to your suggestion. Please see it.

Point 4: There are numerous typos and ambiguous descriptions. Authors should have looked at the manuscripts very carefully before submission. Listed here are only minor examples. There must be many others.

families “based on” should be “in.” I would say all the gene families are formed by gene duplications. I would say, “Gene expression analysis shows the tissue-specific pattern of TPS and TPP genes,”---- Yeast is not a prokaryote. Do you mean “higher plants”? I would say, “For the phylogenetic analyses, 150 full-length BdTPS and BdTPP protein sequences from (insert species names here) were aligned using the ClustalX2.11 program.

Response 3: We are very sorry for the errors in our writing. I checked the manuscript very carefully, and corrected many of the typos and ambiguous descriptions. In addition, I used a professional English editing service to check the manuscript. In fact, I have used this service before submission. Please see the revised manuscript. I  have corrected the errors you mentioned as follows.

families “based on” should be “in.”

Thank you. I checked the manuscript carefully and replaced the ‘based on’ with ‘in’.

I would say all the gene families are formed by gene duplications.

I am very sorry. If you mean that ‘were formed’ should be replaced with ‘ are formed’, I have made this correction.

I would say, “Gene expression analysis shows the tissue-specific pattern of TPS and TPP genes,”

Thank you very much. Because the abstract has a maximum word count limit of approximately 200 words, I deleted this sentences from the abstract.

Yeast is not a prokaryote. Do you mean “higher plants”?

Thank you very much. I have replaced the word ‘prokaryote’ with ‘microorganism’.

I would say, “For the phylogenetic analyses, 150 full-length BdTPS and BdTPP protein sequences from (insert species names here) were aligned using the ClustalX2.11 program.

I modified this sentence to read, ‘For the phylogenetic analyses, full-length BdTPS and BdTPP protein sequences from B. distachyon were aligned using ClustalX2.11,’ please see the revised version.

In addition to the errors you mentioned above, I also found and correct some spelling mistakes. Please see the revised manuscript.

Point 4: 219-222. There is a redundant description.

Response 4: Thank you for your comment. I have deleted the redundant sentence. Please see the revised text.

Point 5: 239-240. Another redundant description.

Response 5: Thank you for your comment. I have deleted the redundant description. Please see the revised text.

Point 6: found, not founded. is extrons a correct term? You don’t need “it was.” Identities, not identifies. That and? noted, you need to use something else. we searched for segmental duplication blocks ----. You need to explain how segmental duplications are defined in this search engine. because, to call tandem duplication, there should be less than five genes between these two genes. You can still call them intrachromosomal duplication or local duplication events. I think the “principle” authors employed is meaningless. This is interchromosomal duplication. You don’t need to regret. indicated or identified. Which one? low levels extra space found, not founded but not in class 1 gene

Response 6:                         

found, not founded.

Thank you. I have replaced ‘founded’ with ‘found’.

is extrons a correct term?

I replaced ‘extrons’ with ‘exons’.

You don’t need “it was.”

I deleted ‘it was’.

Identities, not identifies.

Thank you for this helpful comment. I have replaced ‘identifies’ with ‘identities’.

That and?

I deleted ‘and’

noted, you need to use something else.

I replaced ‘noted’ with ‘showed’

we searched for segmental duplication blocks ----. You need to explain how segmental duplications are defined in this search engine. because, to call tandem duplication, there should be less than five genes between these two genes. You can still call them intrachromosomal duplication or local duplication events. I think the “principle” authors employed is meaningless. This is interchromosomal duplication. You don’t need to regret.

Thanks for your professional comment. I realized that the conclusion regarding ‘segmental duplication’ is not rigorous, and our description is not very accurate. I am sorry about that. According to our current experimental results, we can infer only that these gene pairs may develop from duplication rather than segmental duplication. Therefore, I changed my conclusion. As for you mentioned that ‘the principle is meaningless’, I should explain that I made this evaluation based on some previous research (DOI: 10.1186/s12863-016-0360-y; 10.1093/oxfordjournals.molbev.a004079; 10.1007/s00438-008-0355-0). I think it is the basic criteria for ‘duplication’. I completely agree with your point that ‘This is interchromosomal duplication’. I think our views are not contradictory. The mistake is that I stated that this phenomenon is segmental duplication. I have deleted all references to ‘segmental duplication’.

indicated or identified. Which one?

This word should be ‘indicated’, I deleted ‘identified’.

low levels extra space found, not founded but not in class 1 gene

Thank you. I checked the manuscript carefully and corrected these mistakes.

Point 7: 481-483. Unclear. Consider rewriting this part.

Response 7: Thank you. I have rewritten these two sentences. ‘In contrast, BdTPPs have similar characteristics, suggesting they might have similar gene functions’.

Point 8: 488-490. by comparing the motif sequences ---, we found that motif 7 was specific to BdTPS whereas motif 10 was specific to BdTPP.

Response 8: Thank you very much. I made the suggestion changed.

Point 9: 504-505. This argument is very unlikely.

Response 9: Thank you for this suggestion. This point is the same as Point 1. I have revised this argument according to your comment.

Point 10: chilling tolerance in rice. reconsider the writing of this sentence. more than one?

Response 10: I have rewritten this sentence. ‘OsMAPK3 was found to phosphorylate OsICE1, inhibit its ubiquitination to activate the expression of the OsTPP1 gene, and improve the chilling tolerance of rice.’ Please see the revised texts.

Reviewer 2 Report

Dear Authors,

Genome-wide identification, evolution and expression analysis of trehalose-6-phosphate synthase and trehalose-6-phosphate phosphatase gene famillies in Brachypodium distachyon  

Song Wang, Kai Ouyang, and Kai Wang

I recommend the above manucript for acceptance after minor revision according to the below list.

In general:

the authors should use line spacing 1 in the whole text the species name shall have to put into italics everywhere in the text the References shall have to carefully revise according to the Instruction for the Authors /Plants/MDPI. There are a lot of mistakes in the citation of the authors names the MatMethods shall have to put after the Discussion

Detailed erratum:

Title:

line 3: from instruction for Authors in Plants/MDPI: „When gene or protein names are included, the abbreviated name rather than full name should be used.”

line 4: in title: right - gene families (instead of gene famillies)

Abbreviations:

line 18: in Abbreviatons - the authors should include the Bd (Brachypodium distachyon) as an abbreviation also

Abstract:

line 22: the abstract should be a total of about 200 words maximum (from Instructions for the Authors in Plants/MDPI). Here the abstracts is 240 words, as a consequence the authors should shorten this part.

line 38: may better to write - to one stress or ABA treatments

Introduction:

line 61: delete (.) - does not need the dot

line 63: may better to write - in higher plants

page 76: the authors names are cited wrong in Ref. 21 (line 639). The right citation is the following:

Avonce, N.; Wuyts, J.; Verschooten, K.; Vandesteene, L.; Van Dijck, P.  etc..

line 642 and line 644: Lunn, J.E. is the right citation in Refs. 22-23.

line 646: the authors names are cited wrong in Ref. 24 (line  646). The right citation is the following:

Vandesteene, L., López-Galvis, L.; Vanneste, K.; Feil, R.; Maere, S.; Lammens, W.; Rolland, F.; Lunn, J.E.; Avonce, N.; Beeckman, T.; Van Dijck, P.

line 79:  in Ref 26. the name of the authors are cited wrong. The right citation is the following:

Leyman,  B.; Van Dijck,  P.; Thevelein , J.M.

Line 84: from Ref .28. a part of the authors are missing (line 655), please, correcet like this:

Figueroa, C. M.; Feil, R.; Ishihara, H.; Watanabe, M.; Kölling, K.; Krause, U.; Höhne, M.; Encke, B.; Plaxton, W. C.; Zeeman, S.;  Li, Z.;  Schulze, W.X.;   Hoefgen, R.;  Stitt, M.;   Lunn, J.E.

line 84: from Ref. 29. the last author is missing (line 658), please, correct like this:

Yadav, U. P.; Ivakov, A.; Feil, R.; Duan, G. Y.; Walther, D.; Giavalisco, P.; Piques, M.; Carillo, P.; Hubberten, H. M.; Stitt, M.; Lunn, J.E.

line 86:  the authors names are cited wrong in Ref. 30 (line  662). The right citation is the following:

Kolbe, A.; Tiessen, A.; Schluepmann, H.; Paul, M.;  Ulrich, S.; Geigenberger, P.

line 88: the genetic manipulation of  TPS/TPP genes…

line 89: the authors names are cited wrong in Ref. 37 (line  686). The right citation is the following:

Jang, I-C.; Oh, S.J.; Seo, J.S.; Choi, W.B.; Song, S.I.; Kim, C.H.; Kim, Y.S.; Seo, H.S.; Choi, Y.D.; Nahm, B.H.; Kim, J.K.

line 89: the authors names are cited wrong in Ref. 38 (line  691). The right citation is the following:

Miranda J.A.;   Avonce, N .; Suárez, R .;  Thevelein, J.M  .; Van Dijck, P .; Iturriaga, G.  

line 89: the right author description in Ref. 39.:

Romero, C.; Bellés, J. M.; Vayá, J. L.; Serrano, R., Culiánez-Macià, F. A.

line 91: the authors names are cited wrong in Ref. 40 (line  697). The right citation is the following:

Avonce, N.; Leyman, B.; Mascorro-Gallardo, J.O.; Van Dijck, P.; Thevelein, J.M.; Iturriaga, G.

line 91: the authors names are cited wrong in Ref. 41 (line  700). The right citation is the following:

Nuccio, M. L.; Wu, J.;  Mowers, R.;  Zhou, H.P.;  Meghji, M.;  Primavesi, L.F.; Paul, M.J.;  Chen, X.; Gao, Y.; Haque, E., Basu, S.S.; Lagrimini, L.M.

line 98: the authors should correct the Ref.42 (line 704) like this:

Brkljacic, J.; Grotewold, E.; Scholl, R.; Mockler, T.; Garvin, D. F.; Vain, P.; Brutnell, T.; Sibout, R.; Bevan, M.; Budak, H.; Caicedo, A. L.; Gao, C.; Gu, Y.; Hazen, S. P.; Holt, B. F., 3rd; Hong, S. Y.; Jordan, M.; Manzaneda, A. J.; Mitchell-Olds, T.; Mochida, K.; Mur, L. A.J. ; Park, C. M.; Sedbrook, J.; Watt, M.; Zheng, S. J.; Vogel, J. P. Brachypodium as a model for the grasses: today and the future. Plant Physiology 2011, 157 (1), 3-13.

MatMethds:

line 108: the MatMeth shall have to put after the Discussion according to the Instructions for the Authors in Plants/MDPI.

line 156: The chromosomal distribution was performed with the TBtools software package 155 (http://www.tbtools.com/) (Chen et al., 2018). This reference number should be placed in square brackets [ ] and put into the References.

line 166: reference for the details of sterilization/germination of Brachypodium seeds are necessary here (e.g. Liu et al. 2018).

lines 173-174:  PEG6000 and ABA - where they are arised from: firm name, city, country  names are missing, may be they shall have to put into abbreviations

line 174: how did you stimulate the drought stress, how did you performed the PEG treatment? Please, describe it.

lines 177-179: „Total RNA samples were extracted using a HiPure Plant RNA Min Kit (Magene, city, China) and cDNA synthesis was performed using StarScript II (GeneStar, city, China).” The bolded words shall have to replace.

line 179: „Specific primer pairs and reference genes were designed by qPrimerDB and the ICG database, respectively [54, 55], and are listed in Table S1.” Please, indicate which gene is used as an endogenous control for qRT-PCR here and in the Table S1. Please, also indicate that how many reactions were made (triplicates?) and how many independent biological repetitions were performed for qRT-PCR.

line 181: „using a CFX96 Real-Time PCR Detection System (Bio-Rad, city, USA) under”…

Results:

line 192: „genes was less than those in Arabidopsis thaliana (11), rice (Orysa sativa; 11), Populus trichocarpa”…

line 196: the authors shall have to give a title for the Table 1. After that you may list the abbreviations included in the Table 1. Indicate in the abbreviations that what CDS and pI mean (protein lenght in bp and isoelectric point).

lines 216- 215: „phylogenetic tree was constructed using the complete TPS/TPP protein sequences with the MEGA 7.0  program []. Reference is missing here.

line 219:  Correct the  Ref. 58. (line 754) like this:

Poueymiro, M.; Cazalé, A. C.; Francois, J. M.; Parrou, J. L.; Peeters, N.; Genin, S.  A Ralstonia solanacearum type III effector directs the production of the plant signal metabolite trehalose-6-phosphate. mBio 2014, 5 (6), e02065-14.

line 219: Correct the authors name in Ref. 59 (line 757):

Van Dijck,  P.; Mascorro-Gallardo,  J.O.; De Bus, M.; Royackers, K.; Iturriaga, G.;, Thevelein, J.M.

lines 239-241: these sentences are not understandable

line 241: „All TPP genes have similar 241 extrons (6-10), „  - exons?

line 271: „In addition, to compare with the differences in of TPP domains between BdTPS”

line 274: „amino oacid identity on average”

lines 281-283: „First, the Class II of BdTPSs possessesd nine motifs that and were lacked motif 10.”  These sentences are not understandabe, plese, correct them.

line 293: „from the B. distachyong

lines 303-304: „red dotted lines represent” – these dotted lines are very difficult to see, please somehow change this for anything else. The dot is missing from the end of the sentence.

line 325: a reference is important for the explanation of the meaning of Ka/Ks values

line 325: „which was similar to the results obtained in other plants”

line 328:  the title of Table 2 should put under the Table2 (into line 329) and in the Table 2 legend it is important to explain better the abbreviations, eg. Ka/Ks values.

line 333: „The results are listed in Table 2 and showed what shows that”

line 347: the authors names are cited wrong in Ref. 63 (line 770). The right citation is the following:         van Dijken, A.J.H.; Schluepmann, H.; Smeekens, S.C.M.

line 354: TC-rich repeats – what do they are good for?

line 355: again: LTRs and  TGACG motifs – how do they relate to abiotic stresses?

line 360: Ref. 66 (line 779)  is from a book, please, cite it like a book chapter.

line 362:  in  Ref.69. (line 787) the last author name is wrong, the right one is the next:

Zhang, T.; Tan, D.; Li, Z.; Zhang, X.; Han, Z.

line 426. dot is missing from the end of the sentence

lines 428- 429: „To further investigate the functions of BdTPS and BdTPP genes in response to abiotic stresses,”  - you talked before always about abiotic stresses and cold, salinity and drought stresses are considered as abiotic stresses

line 448: „Figure 7. Expression patterns of TPS and TPP gene family under cold, salt and drought stress conditions”

line 448: in Figure 7. it should be better to nominate the control as CT (not CK)

line 442: „In this study, we also examined the gene expression pattern”

Discussion:

lines 456-458: „The TPS family has been reported in many species, such as Arabidopsis, Oryza sativa, Nelumbo nucifera, Solanum tuberosum, Moringa oleifera, Gossypium …. and Triticum aestivum [8, 26, 44-47, 71]. Please, unify this sentence for species names…

lines 466-468: „All of these BdTPS genes contained Glyco-transf-20 (TPS) and Trehalose-PPase (TPP) domains, and all of the BdTPP genes included only Trehalose-PPase domains.”

line 487: „However, we also found that the Trehalose-PPase domains have…”

line 516: „stress tolerance in different plant species”

line 524: „better understanding the abotic stress-related transduction pathways in B. distachyon.”

line 540: „has been shown in rice OsTPS1 [Reference ].”

line 560: „these genes were present in all tissues at various expression levels”.

References:

- correct the References according to the Instructions for the Authors in Plants/MDPI

-the species name shall have to put in italics in every reference

- volumes shall have to put in italics in every reference

- page range shall have to indicate where it is available

Sincerely yours,

Reviewer 1

Author Response

Dear editior and Reviewers,

Thank you very much. This comments are very valuable and very helpful for revising and improving our paper, as well as the important guiding significance to our researches. We have studied comments carefully and have made correction which we hope meet with approval. The main corrections in the paper and the responds to the reviewer’s comments are as following:

Point 1:

In general:

the authors should use line spacing 1 in the whole text the species name shall have to put into italics everywhere in the text the References shall have to carefully revise according to the Instruction for the Authors /Plants/MDPI. There are a lot of mistakes in the citation of the authors names the MatMethods shall have to put after the Discussion

Detailed erratum:

Title:

line 3: from instruction for Authors in Plants/MDPI: „When gene or protein names are included, the abbreviated name rather than full name should be used.”

Response 1: Thank you for your suggestion. I have adjusted the line spacing to 1 in the whole text; used italics to mark the species names, and altered the citation format based on the Instruction for the Authors /Plants/MDPI.

I have used abbreviated names to replace the full names of genes.

Point 2: line 4: in title: right - gene families (instead of gene famillies)

Response 2: According to your suggestion, we have corrected ‘famillies’ to ‘families’.

Point 3: Abbreviations: line 18: in Abbreviatons - the authors should include the Bd (Brachypodium distachyon) as an abbreviation also

Response 3: Thank you for this helpful suggestion. I have added the Bd, Brachypodium distachyon to the Abbreviations list.

Point 4: Abstract: line 22: the abstract should be a total of about 200 words maximum (from Instructions for the Authors in Plants/MDPI). Here the abstracts is 240 words, as a consequence the authors should shorten this part.

Response 4: Thank you for this suggestion. I deleted some unimportant sentences and modified other sentences to bring the total number of words within 200.

Point 5: line 38: may better to write - to one stress or ABA treatments

Response 5: Thank you for making this point. We have corrected the sentence based on your suggestion.

Point 6: Introduction: line 61: delete (.) - does not need the dot

Response 6: Thank you for your carefully attention, we have deleted this dot.

Point 7: line 63: may better to write - in higher plants

Response 7: Thank you for this suggestion. We have replaced ‘in high plants’ with ‘in higher plants’.

Point 8: page 76: the authors names are cited wrong in Ref. 21 (line 639). The right citation is the following:

Avonce, N.; Wuyts, J.; Verschooten, K.; Vandesteene, L.; Van Dijck, P.  etc..

Response 8: Thank you very much. We have corrected the reference format of Ref. 21; please see the revised manuscript.

Point 9: line 642 and line 644: Lunn, J.E. is the right citation in Refs. 22-23.

Response 9: Thank you again. I have replaced the Lunn.J with Lunn,J.E.

Point 10: line 646: the authors names are cited wrong in Ref. 24 (line  646). The right citation is the following:

Vandesteene, L., López-Galvis, L.; Vanneste, K.; Feil, R.; Maere, S.; Lammens, W.; Rolland, F.; Lunn, J.E.; Avonce, N.; Beeckman, T.; Van Dijck, P.

Response 10: We have corrected the format of Ref. 24; please see the revised manuscript.

Point 11: line 79:  in Ref 26. the name of the authors are cited wrong. The right citation is the following:

Leyman,  B.; Van Dijck,  P.; Thevelein , J.M.

Response 11: We have corrected the format of Ref. 26; please see the revised manuscript.

Point 12: Line 84: from Ref .28. a part of the authors are missing (line 655), please, correcet like this:

Figueroa, C. M.; Feil, R.; Ishihara, H.; Watanabe, M.; Kölling, K.; Krause, U.; Höhne, M.; Encke, B.; Plaxton, W. C.; Zeeman, S.;  Li, Z.;  Schulze, W.X.;   Hoefgen, R.;  Stitt, M.;   Lunn, J.E.

Response 12: We have corrected the format of Ref. 28; please see the revised manuscript.

Point 13: line 84: from Ref. 29. the last author is missing (line 658), please, correct like this:

Yadav, U. P.; Ivakov, A.; Feil, R.; Duan, G. Y.; Walther, D.; Giavalisco, P.; Piques, M.; Carillo, P.; Hubberten, H. M.; Stitt, M.; Lunn, J.E.

Response 13: We have corrected the format of Ref. 29 and added the last author; please see the revised manuscript.

Point 14: line 86:  the authors names are cited wrong in Ref. 30 (line  662). The right citation is the following:

Kolbe, A.; Tiessen, A.; Schluepmann, H.; Paul, M.; Ulrich, S.; Geigenberger, P.

Response 14: We have corrected the format of Ref. 30; please see the revised manuscript.

Point 15: line 88: the genetic manipulation of TPS/TPP genes…

Response 15: Thank you. We have added the word ‘of’ to this sentence.

Point 16: line 89: the authors names are cited wrong in Ref. 37 (line 686). The right citation is the following:

Jang, I-C.; Oh, S.J.; Seo, J.S.; Choi, W.B.; Song, S.I.; Kim, C.H.; Kim, Y.S.; Seo, H.S.; Choi, Y.D.; Nahm, B.H.; Kim, J.K.

Response 16: We have corrected the author names in Ref. 37; please see the revised manuscript.

Point 17: line 89: the authors names are cited wrong in Ref. 38 (line 691). The right citation is the following:

Miranda J.A.; Avonce, N .; Suárez, R .; Thevelein, J.M .; Van Dijck, P .; Iturriaga, G. 

Response 17: We have corrected the author names in Ref. 38; please see the revised manuscript.

Point 18: line 89: the right author description in Ref. 39.:

Romero, C.; Bellés, J. M.; Vayá, J. L.; Serrano, R., Culiánez-Macià, F. A.

Response 18: We have corrected the author names of Ref. 39; please see the revised manuscript.

Point 19: line 91: the authors names are cited wrong in Ref. 40 (line 697). The right citation is the following:

Avonce, N.; Leyman, B.; Mascorro-Gallardo, J.O.; Van Dijck, P.; Thevelein, J.M.; Iturriaga, G.

Response 19: We have corrected the author names in Ref. 40; please see the revised manuscript.

Point 20: line 91: the authors names are cited wrong in Ref. 41 (line  700). The right citation is the following:

Nuccio, M. L.; Wu, J.;  Mowers, R.;  Zhou, H.P.;  Meghji, M.;  Primavesi, L.F.; Paul, M.J.;  Chen, X.; Gao, Y.; Haque, E., Basu, S.S.; Lagrimini, L.M.

Response 20: We have corrected the author names in Ref. 41; please see the revised manuscript.

Point 21: line 98: the authors should correct the Ref.42 (line 704) like this:

Brkljacic, J.; Grotewold, E.; Scholl, R.; Mockler, T.; Garvin, D. F.; Vain, P.; Brutnell, T.; Sibout, R.; Bevan, M.; Budak, H.; Caicedo, A. L.; Gao, C.; Gu, Y.; Hazen, S. P.; Holt, B. F., 3rd; Hong, S. Y.; Jordan, M.; Manzaneda, A. J.; Mitchell-Olds, T.; Mochida, K.; Mur, L. A.J. ; Park, C. M.; Sedbrook, J.; Watt, M.; Zheng, S. J.; Vogel, J. P. Brachypodium as a model for the grasses: today and the future. Plant Physiology 2011, 157 (1), 3-13.

Response 21: We have corrected the author names of Ref. 42; please see the revised manuscript.

Point 22: MatMethds: line 108: the MatMeth shall have to put after the Discussion according to the Instructions for the Authors in Plants/MDPI.

Response 22: Thank you for this suggestion. I checked the format of ‘Plants’, and adjusted the layout of the manuscript. Please see the revised manuscript.

Point 23: line 156: The chromosomal distribution was performed with the TBtools software package 155 (http://www.tbtools.com/) (Chen et al., 2018). This reference number should be placed in square brackets [ ] and put into the References.

Response 23: Thank you for this suggestion. I have added the reference as suggested.

Point 24: line 166: reference for the details of sterilization/germination of Brachypodium seeds are necessary here (e.g. Liu et al. 2018).

Response 24: Thank you for this suggestion. I have added the reference as suggested.

Point 25: lines 173-174:  PEG6000 and ABA - where they are arised from: firm name, city, country names are missing, may be they shall have to put into abbreviations

Response 25: I have added this missing information. PEG 6000, Coolaber, Beijing, China. ABA, Solarbio, Beijing, China.

Point 26: line 174: how did you stimulate the drought stress, how did you performed the PEG treatment? Please, describe it.

Response 26: Thank you for your suggesting. However, this is a general approach in which 20% (w/v) PEG 6000 solution is used to simulate moderate drought. Many studies have used this method. The seedling is simply moved to this solution for different lengths of time, as in the case of salt stress. Please consider this point.

Point 27: lines 177-179: Total RNA samples were extracted using a HiPure Plant RNA Min Kit (Magene, city, China) and cDNA synthesis was performed using StarScript II (GeneStar, city, China).” The bolded words shall have to replace.

Response 27: Thank you for the suggestion. I have added city names in this sentence, such as Magene, Guangzhou, China and GeneStar, Beijing, China.

Point 28: line 179: Specific primer pairs and reference genes were designed by qPrimerDB and the ICG database, respectively [54, 55], and are listed in Table S1.” Please, indicate which gene is used as an endogenous control for qRT-PCR here and in the Table S1. Please, also indicate that how many reactions were made (triplicates?) and how many independent biological repetitions were performed for qRT-PCR.

Response 29: Thanks for this helpful advise. The reference gene was named SamDC. Four reactions were peferened and 3 independent biological repetitions were performed for qRT-PCR. These changes have been made to the manuscript, please see the revised text.

Point 29: line 181: „using a CFX96 Real-Time PCR Detection System (Bio-Rad, city, USA) under”…

Response 29: We have added the city name ‘Hercules’ to this sentence.

Point 30: Results: line 192: „genes was less than those in Arabidopsis thaliana (11), rice (Orysa sativa; 11), Populus trichocarpa”…

Response 30: Thank you , we have added the Oryza sativa to this sentence.

Point 31: line 196: the authors shall have to give a title for the Table 1. After that you may list the abbreviations included in the Table 1. Indicate in the abbreviations that what CDS and pI mean (protein lenght in bp and isoelectric point).

Response 31: We are very sorry for our neglecting the title. We have added the title name that ‘Table 1 Basic characteristic of TPS and TPP genes in Brachypodium distachyon’. In addition, we also added definition of the abbreviations CDS and pI (protein lenght in bp and isoelectric point).

Point 32: lines 216- 215: „phylogenetic tree was constructed using the complete TPS/TPP protein sequences with the MEGA 7.0 program []. Reference is missing here.

Response: Thank you for this point. This reference (Ref.50) is provided in the materials and methods section.

Point 33: line 219:  Correct the  Ref. 58. (line 754) like this:

Poueymiro, M.; Cazalé, A. C.; Francois, J. M.; Parrou, J. L.; Peeters, N.; Genin, S.  A Ralstonia solanacearum type III effector directs the production of the plant signal metabolite trehalose-6-phosphate. mBio 2014, 5 (6), e02065-14.

Response 33: Thank you for this suggestion. We have corrected the format based on your suggestion.

Point 34: line 219: Correct the authors name in Ref. 59 (line 757):

Van Dijck,  P.; Mascorro-Gallardo,  J.O.; De Bus, M.; Royackers, K.; Iturriaga, G.;, Thevelein, J.M.

Response 34: We have corrected the author names in Ref.59.

Point 35: lines 239-241: these sentences are not understandable

Response 35: There are two sentences on lines 239-241. That is ‘In contrast to the similar CDS length (2463-2958) among the nine TPS genes, it is extremely variable on genomic length was extremely variable (2783-7172). However, the same phenomenon was not been founded in TPPs.’ What I wanted to express is that TPS genes similar CDS lengths, but vary greatly in genomic length; however, in TPP genes, both CDS and genomic length are similar. I have rewritten these two sentences.

Point 36: line 241: „All TPP genes have similar 241 extrons (6-10), „  - exons?

Response 36: Thank you for this suggestion. Yes, this word should be exons, and I have made this correction.

Point 37: line 271: „In addition, to compare with the differences in of TPP domains between BdTPS”

Response 37: I have deleted ‘with’ and ‘of ’, thank you for this comment.

Point 38: line 274: „amino oacid identity on average”

Response 38: I have corrected ‘oacid’ to ‘acid’.

Point 39: lines 281-283: „First, the Class II of BdTPSs possessesd nine motifs that and were lacked motif 10.”  These sentences are not understandabe, plese, correct them.

Response 39: I have rewritten this sentence. First, all BdTPS proteins of Class II contained nine motifs excluding motif 10.

Point 40: line 293: „from the B. distachyong”

Response 40: I have corrected ‘B. distachyong’ to ‘B. distachyon’

Point 41: lines 303-304: „red dotted lines represent” – these dotted lines are very difficult to see, please somehow change this for anything else. The dot is missing from the end of the sentence.

Response 41: Thank you for your comment. I have re-drawn this figure, and used a full line to replace dotted line. Please see the revised version.

Point 42: line 325: a reference is important for the explanation of the meaning of Ka/Ks values

Response 42: I have added the reference as suggested.

Point 43: line 325: „which was similar to the results obtained in other plants”

Response 43: I have added the word ‘obtained’ in this sentence.

Point 44: line 328:  the title of Table 2 should put under the Table2 (into line 329) and in the Table 2 legend it is important to explain better the abbreviations, eg. Ka/Ks values.

Response 44: Thank you for your comment. First, compared with published articles in ‘Plants’ and other journal, I think most table titles should be at the top rather than the bottom, whereas the titles of figures are generally placed at the bottom. Second, I have add the explanation of Ka and Ks to the legend. Please see the revised version.

Point 45: line 333: „The results are listed in Table 2 and showed what shows that”

Response 45: Thank you for you comment, I have replaced the ‘and showed’ with ‘what shows ’.

Point 46: line 347: the authors names are cited wrong in Ref. 63 (line 770). The right citation is the following:      

Van Dijken, A.J.H.; Schluepmann, H.; Smeekens, S.C.M.

Response 46: We have corrected the author names in Ref.63.

Point 47: line 354: TC-rich repeats – what do they are good for?

Response 47: TC-rich repeats are a stress response element. I have added the explanation.

Point 48: line 355: again: LTRs and TGACG motifs – how do they relate to abiotic stresses?

Response 48: LTRs and TGACG motifs are also stress response elements. I have added some information. In addition, we have introduced the subject in Materials and methods section. Please see the revised version.

Point 49: line 360: Ref. 66 (line 779) is from a book, please, cite it like a book chapter.

Response 49: I have corrected this format of this citation.

Point 50: line 362:  in  Ref.69. (line 787) the last author name is wrong, the right one is the next:

Zhang, T.; Tan, D.; Li, Z.; Zhang, X.; Han, Z.

Response 50: We have corrected the author names in Ref.69.

Point 51: line 426. dot is missing from the end of the sentence

Response 51: Thank you. I have added the dot at the end of the sentence.

Point 52: lines 428- 429: „To further investigate the functions of BdTPS and BdTPP genes in response to abiotic stresses,”  - you talked before always about abiotic stresses and cold, salinity and drought stresses are considered as abiotic stresses

Response 52: Thank you. I have corrected this careless error.

Point 53: line 448: „Figure 7. Expression patterns of TPS and TPP gene family under cold, salt and drought stress conditions”

Response 53: I have revised this figure caption.

Point 54: line 448: in Figure 7. it should be better to nominate the control as CT (not CK)

Response 54: I have replaced CK with CT in Fig.7.

Point 55: line 442: „In this study, we also examined the gene expression pattern”

Response 55: I have added the ‘gene’ in this sentence.

Point 56: Discussion: lines 456-458: „The TPS family has been reported in many species, such as Arabidopsis, Oryza sativa, Nelumbo nucifera, Solanum tuberosum, Moringa oleifera, Gossypium …. and Triticum aestivum [8, 26, 44-47, 71]. Please, unify this sentence for species names…

Response 56: Thank you very much. I have made the species names consistent. Please see the revised version.

Point 57: lines 466-468: „All of these BdTPS genes contained Glyco-transf-20 (TPS) and Trehalose-PPase (TPP) domains, and all of the BdTPP genes included only Trehalose-PPase domains.”

Response 57: Thank you for your suggestion. Nevertheless, I prefer to use Glyco_transf_20, rather than Glyco-transf-20 as the former, is the standard name. Please see http://pfam.xfam.org/family/PF00982 or some articles such as ‘https://doi.org/10.1007/s00425-018-2945-3’.

Point 58: line 487: „However, we also found that the Trehalose-PPase domains have…”

Response 58: Thank you for your suggestion. Please see Point 57 above, as the reply to this point is the same.

Point 59: line 516: „stress tolerance in different plant species”

Response 59: I have made the suggested correction.

Point 60: line 524: „better understanding the abotic stress-related transduction pathways in B. distachyon.”

Response 60: Thank you. I have add ‘abiotic stress-related’ in this sentence.

Point 61: line 540: „has been shown in rice OsTPS1 [Reference ].”

Response 61: I have added the reference.

Point 62: line 560: „these genes were present in all tissues at various expression levels”.

Response 62: Thank you for this suggestion. I have replaced ‘involved’ with ‘present’.

Point 63: References: correct the References according to the Instructions for the Authors in Plants/MDPI

Response 63: I have corrected the reference formation based on the Instructions for ‘Plants’.

Point 64: the species name shall have to put in italics in every reference

Response 64: Thank you. I have corrected the format. Please see the revised version

Point 65: volumes shall have to put in italics in every reference

Response 65: Thank you for this suggestion. I checked every reference and put volume numbers in italics.

Point 66: page range shall have to indicate where it is available

Response 66: Thank you for this suggestion. I have added the page range to the references.

Reviewer 3 Report

Some paragraphs from the results section may be more appropriate to include in the discussion section.

lines 236-238 “Previous theories…..BdTPSgenes”

Lines 245-248 “This phenomenom…..gene in plants”

Line 293 B. distachyong. Please correct the name

Line 398 Figure 5 B should be redrafted to achieve a higher definition.

Line 457 Solanum tuberosum two words, no Solanumtuberosum

Line 631 6091-6100, Please correct number of the last page

Line 634 Please correct number of the last page

Line 638 Please correct number of last the page

Line 648 Please check and correct the initial and final pages of the article. Plant Physiology 2012, 160 (2) 884-896

Line 672 Please correct number of the last page

Line 675 Please correct number of the last page

Line 679 Please correct number of the last page

Line 690 Please check and correct the initial and final pages of the article. Plant physiology, 2003, vol. 131, no 2, p. 516-524.

Line 719 Please Check the reference BMC Genetics 2016, 17 (1), 1-1

Line 734 Please correct number of the last page

Line 753 Please correct number of the last page

Line 775 Please correct number of the last page

Line 779. Reference 66 cite the reference correctly Volume, issue, pages…

Author Response

Point 1: Some paragraphs from the results section may be more appropriate to include in the discussion section.

lines 236-238 “Previous theories…..BdTPS genes”

Lines 245-248 “This phenomenom…..gene in plants”

Response 1: Thank you for your suggestions. We have moved the two sentences that you mentioned from the results section to the discussion section. Please see the revised manuscript.

Point 2

Line 293 B. distachyong. Please correct the name

Response 2: Thank you for this suggestion. We corrected ‘B. distachyong’ to ‘B. distachyon’.

Point 3

Line 398 Figure 5 B should be redrafted to achieve a higher definition.

Response 3: Thank you for your suggestions. We have corrected the definition of Figure 5B. In addition, I splited the Fig 5 as Fig5, 6, 7.

Point 4: Line 457 Solanum tuberosum two words, no Solanumtuberosum

Response 4: Thank you for this suggestion. We corrected ‘Solanumtuberosum’ to ‘Solanum tuberosum’.

Point 5:Line 631 6091-6100, Please correct number of the last page

Response 5: I have added the last page in this citation.

Point 6: Line 634 Please correct number of the last page

Response6: Thank you for this suggestion. I have added the last page to this citation.

Point 7: Line 638 Please correct number of last the page

Response7: I have added the last page to this citation.

Point 8: Line 648 Please check and correct the initial and final pages of the article. Plant Physiology 2012, 160 (2) 884-896

Response 8: Thank you for this suggestion. I have added the initial and final pages to this citation.

Point 9: Line 672 Please correct number of the last page

Response 9: Thank you for this suggestion. I have added the last page to this citation.

Point 10: Line 675 Please correct number of the last page

Response 10: Thank you for this suggestion. I have added the last page to this citation.

Point 11: Line 679 Please correct number of the last page

Response 11: Thank you for this suggestion. I have added the last page to this citation.

Point 12: Line 690 Please check and correct the initial and final pages of the article. Plant physiology, 2003, vol. 131, no 2, p. 516-524.

Response: Thank you for this suggestion. I have checked this citation, and the initial and final pages are 516-524.

Point 13: Line 719 Please Check the reference BMC Genetics 2016, 17 (1), 1-1

Response13: Thank you for this suggestion. I have corrected the last page to this citation. Please see the revised text.

Point 14: Line 734 Please correct number of the last page

Response 14: Thank you for this suggestion. I have added the last page to this citation.

Point 15: Line 753 Please correct number of the last page

Response 15: Thank you for this suggestion. I have added the last page to this citation.

Point 16: Line 775 Please correct number of the last page

Response 16: Thank you for this suggestion. I have added the last page to this citation.

Point 17: Line 779. Reference 66 cite the reference correctly Volume, issue, pages…

Response 17: Thank you for this suggestion. Ref. 66 is from a book, and I have added some information on this reference.

Finally, I checked all the references based on the formatting instructions for ‘Plants’ and made appropriate changes.

Round 2

Reviewer 1 Report

In Figure 10, there are "a", "b" and "c" in the histograms, but what they mean is not described. Please clarify them in the figure legend.

I would suggest in the abstract that "1 BdTPS gene pair and 4 BdTPP gene pairs are formed by RELATIVELY RECENT duplication events." 

Author Response

Comment 1: In Figure 10, there are "a", "b" and "c" in the histograms, but what they mean is not described. Please clarify them in the figure legend.

Response 1: Thank you for this comment. I have added the description of figure legend to read, ‘Different lowercase letters above the bars indicate significantly different values (P < 0.05).’ Please see the manuscript. 

Comment 2: I would suggest in the abstract that "1 BdTPS gene pair and 4 BdTPP gene pairs are formed by RELATIVELY RECENT duplication events."

Response 2: Thank you very much. I made the suggestion changed.

Reviewer 3 Report

The manuscript has been extensively modified following the reviewers' recommendations. 

Author Response

Thank you very much!